# Discovery of a proteolytic flagellin family in diverse bacterial phyla that assembles enzymatically active flagella

Ulrich Eckhard [1,4], Hina Bandukwala[2], Michael J. Mansfield[2], Giada Marino [1], Jiujun Cheng[2], Iain Wallace[2], Todd Holyoak[2], Trevor C. Charles [2], John Austin[3], Christopher M. Overall [1] & Andrew C. Doxey [2]

Bacterial flagella are cell locomotion and occasional adhesion organelles composed primarily of the polymeric protein flagellin, but to date have not been associated with any enzymatic function. Here, we report the bioinformatics-driven discovery of a class of enzymatic flagellins that assemble to form proteolytically active flagella. Originating by a metallopeptidase insertion into the central flagellin hypervariable region, this flagellin family has expanded to at least 74 bacterial species. In the pathogen, *Clostridium haemolyticum*, metallopeptidase-containing flagellin (which we termed flagellinolysin) is the second most abundant protein in the flagella and is localized to the extracellular flagellar surface. Purified flagellar filaments and recombinant flagellin exhibit proteolytic activity, cleaving nearly 1000 different peptides. With ~20,000 flagellin copies per ~10-μm flagella this assembles the largest proteolytic complex known. Flagellum-mediated extracellular proteolysis expands our understanding of the functional plasticity of bacterial flagella, revealing this family as enzymatic biopolymers that mediate interactions with diverse peptide substrates.

[1] Life Sciences Institute, Department of Oral Biological and Medical Sciences, Faculty of Dentistry, University of British Columbia, 2350 Health Sciences Mall, Vancouver, British Columbia, Canada V6T 1Z3. [2] Department of Biology, University of Waterloo, 200 University Ave. West, Waterloo, Ontario, Canada N2L 3G1. [3] Bureau of Microbial Hazards, Health Products and Food Branch, Health Canada, Ottawa, Ontario, Canada K1A 0K9. [4] Present address: Department of Cell and Molecular Biology, Uppsala University, BMC, Box 596, 751 24 Uppsala, Sweden. Ulrich Eckhard, Hina Bandukwala and Michael J. Mansfield contributed equally to this work. Christopher M. Overall and Andrew C. Doxey jointly supervised this work. Correspondence and requests for materials should be addressed to C.M.O. (email: chris.overall@ubc.ca) or to A.C.D. acdoxey@uwaterloo.ca)

One of the most remarkable molecular structures in nature is the bacterial flagellum: a self-assembling, 10-μm long molecular nanomachine responsible for cell motility[1, 2]. Although primarily associated with locomotion, the roles of flagella have expanded to include a diverse range of biological phenomena including adhesion[2], mechanosensing[3], biofilm interactions[3–5], and virulence[6, 7]. This functional diversity is mirrored at the molecular level, as the genes encoding flagellar proteins and the structure of flagellar machinery can vary considerably between species[8].

The primary structural component of bacterial flagella is the protein flagellin. Up to 20,000 flagellin monomers assemble to form a helical, hollow filament ~ 20 nm in diameter and ~ 10 μm in length[9], which rotates via a proton/sodium motive force to drive cell motility[10]. Thus, any function encoded by a single flagellin monomer can be massively amplified upon filament polymerization. Flagellin monomers are composed of three structural domains: slowly evolving N-terminal and C-terminal coiled-coil domains that interact in cis to form the core of the filament, and a central hypervariable region of extreme sequence variation, which forms the filament surface[8, 11] (Fig. 1a) and thus, is largely responsible for interfacing with the environment[6]. These interactions play an important role in adherence and colonization of host cells in a diverse range of mammalian[12–14] and non-mammalian bacterial pathogens[15, 16]. As a result, flagellins also constitute an important class of pathogen-associated molecular patterns (PAMPs), and are recognized by toll-like receptor 5 in the innate immune system of mammals, flies and plants[17].

Given the extreme sequence diversity of the hypervariable region and its potential as a scaffold for surface-localized functions, we hypothesized that there are flagellins with novel functionality that can be predicted from available bacterial genomes. Here, by computational identification of unexpected domain fusions[18], we report the discovery and experimental validation of a family of enzymatically active flagellins present in the genomes of 74 bacterial species including the pathogenic clostridia *Clostridium haemolyticum*, and strains of *Clostridium novyi* and *Clostridium botulinum*. We find that these flagellins harbor a catalytically active zinc-metallopeptidase domain that is localized to extracellular flagellar filaments, so resulting in flagella-embedded protease activity in structures of up to 10 μm. Flagellin-mediated proteolysis expands our understanding of the functional plasticity of flagellar filaments as enzymatic biopolymers, with potential for numerous roles in saprophytic bacteria and in pathogens including biofilm interactions and colonization, tissue colonization and virulence.

## Results

**Computational prediction of a proteolytic flagellin family.** To explore novel functionality in bacterial flagellins, we gathered all 26,587 predicted flagellin sequences from the NCBI non-redundant database and examined them for novel domain architectures. Sixty-one different domains were identified within the flagellin hypervariable region, revealing tremendous domain diversity associated with surface-exposed structures on the flagellar filament (Fig. 1a). Bacterial flagellins are not known to possess enzymatic activity. Therefore, it was unexpected to identify several putative enzymatic domains within the flagellin hypervariable regions (Table 1, Supplementary Table 1). The most frequently occurring flagellin enzymatic domain identified was the Peptidase M9 family domain, a thermolysin-like glu-zincin metallopeptidase domain found in bacterial collagenases (Fig. 1b) (Table 1). Sequences annotated as Peptidase M26[19]

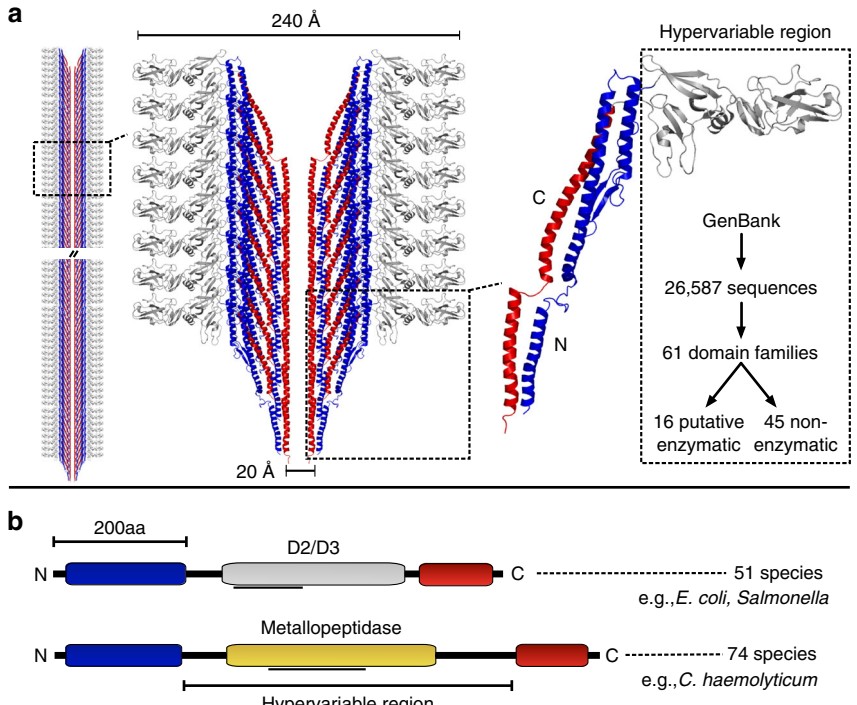

**Fig. 1** Structural model of the flagellar filament and identification of uncharacterized surface-exposed flagellin domains. **a** Structural model of the flagellar filament and constituent flagellin proteins (*left*), highlighting the interior flagellin N-terminal (*blue*) and C-terminal (*red*) domain and the surface-exposed hypervariable region domain (*gray*). The model is based on the structure of FliC from *Salmonella* (PDB 1ucu). Flagellin hypervariable regions from the NCBI GenBank database were analyzed, revealing 61 putative domain families including several with potential enzymatic function. **b** Schematic depiction of the commonly studied D2/D3 flagellin domain variant, as well as the novel putative metallopeptidase domain predicted within 74 species. The *gray lines* below the domains indicate regions matched by the Conserved Domain Database, which were manually refined through subsequent analysis

| Rank | Domain | Frequency |
|---|---|---|
| | **Table 1 Top 10 most frequent protein domains detected within flagellin hypervariable regions according to the NCBI conserved domain database** | |
| 1 | Flagellin_IN | 3286 |
| 2 | Flagellin_D3 | 715 |
| 3 | FliC | 417 |
| 4 | DUF1522 | 155 |
| 5 | *Putative Gluzincin Family Metallopeptidase* | 86 |
| 6 | FliC_SP | 67 |
| 7 | DUF3383 | 24 |
| 8 | vWFA (von Willibrand Factor A) | 21 |
| 9 | *Peptidase_M26* | 17 |
| 10 | *Sialidase* | 10 |
| Putative enzymatic domains are italicized | | |

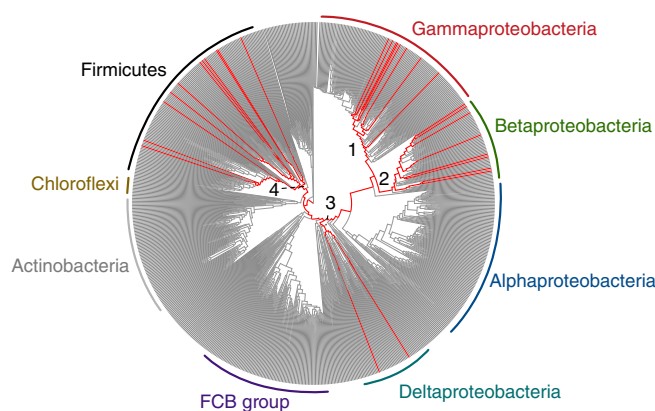

**Fig. 2** Phylogenomic distribution of proteolytic flagellins on bacterial tree of life. Genera with predicted proteolytic flagellin genes are indicated by *red lines*. The presence/absence pattern is highly scattered indicative of extensive lateral gene transfer. Proteolytic flagellins were identified in four major lineages (numbered 1–4). The genus-level bacterial phylogenetic tree was derived from an earlier study[49]

family domains, which include metallopeptidases that cleave mammalian IgA, were also identified within some flagellins (Table 1). After detection of additional homologs via PSI-BLAST, a family of 86 metallopeptidase-containing flagellin sequences were identified from 74 bacterial species and 35 genera (Supplementary Table 2). These species are phylogenetically diverse, indicative of lateral gene transfer, and most commonly occur in the Firmicutes, Betaproteobacteria and Gammaproteobacteria (Fig. 2). In addition, metallopeptidase-containing flagellins are present in several animal pathogens of both medical and agricultural importance including *C. haemolyticum*, *C. novyi* A str. 4570, and *C. botulinum* C/D str. DC5 (Supplementary Table 2). Despite the presence of flagellin and protease domains in all identified members of this flagellin family, most have been labeled as "hypothetical proteins" in the database (Supplementary Table 2), which may have contributed to their previous lack of identification and characterization.

To verify homology of this flagellin hypervariable region family to zinc metallopeptidases, we performed in-depth sequence analysis (Fig. 3a, b) and structural homology modeling (Fig. 3c–e).

Reciprocal PSI-BLAST searches[20] and three independent structure prediction methods[21–23] all identified the peptidase domain of clostridial collagenases as the closest homolog. Notwithstanding a low overall sequence identity of ~13%, we constructed a high-confidence structural model based on the available crystal structures of clostridial collagenases (Fig. 3c), which further guided the identification of key proteolytic motifs. Remarkably, all of the identified flagellins conserve the critical $Zn^{++}$ ion-binding histidine residues in the HExxH catalytic motif—the glutamate being the general base, as well as additional key residues in close proximity to the active site, including the more distal third zinc ion ligating glutamate and Ala265 forming the metalloprotease hydrophobic "basement" (Fig. 3b–d). Thus, members of the domain family are putative gluzincin metallopeptidases[19]. Their functional importance in these flagellins was further indicated by conservation of the enzymatic motifs rather than their loss through pseudogenization. Genomic context analysis also identified flagellar genes and other peptidases as gene neighbors of metallopeptidase-containing flagellins (Supplementary Fig. 1), suggesting that they form both part of the flagellar machinery and are involved in extracellular proteolytic pathways.

**Flagellin from *C. haemolyticum* is an active protease.** We selected the flagellin metallopeptidase protein FliA(H) from the animal pathogen *C. haemolyticum* for biochemical characterization. Notably, *C. haemolyticum* is associated with collagenolytic activity and host-tissue degradation[24]. FliA(H) forms a subfamily

of flagellins containing a metallopeptidase domain within the hypervariable region, and the absence of this domain in related sequences implies an ancestral insertion event (Supplementary Fig. 2). To focus on the putative gluzincin metallopeptidase domain in the absence of flagellin polymerization, we designed a recombinant cDNA construct containing the FliA(H)-hypervariable region only lacking the N- terminal and C-terminal coiled-coil domain. We expressed and purified the putative metallopeptidase domain to high purity (Fig. 4a), without other protease contaminants as verified by liquid chromatography –tandem mass spectrometry (LC–MS/MS) (Supplementary Fig. 3, Supplementary Table 3).

We utilized a high throughput proteomic approach known as proteomic identification of protease cleavage sites (PICS)[25, 26] to confirm and characterize the metallopeptidase activity of the FliA(H) hypervariable region and peptide substrate specificity. PICS profiles both the prime (P′) and non-prime (P) amino acid preferences of cleavage sites using biological-derived database searchable peptide libraries. PICS confirmed FliA(H)-hypervariable region as an active peptidase, cleaving 391 peptides in a trypsin-generated peptide library, with each peptide ending in either an arginine or a lysine residue (Fig. 4b). To exclude potential library bias, we repeated the PICS analysis using a GluC-generated peptide library, with each peptide ending at glutamate or aspartate residues. Again, the FliA(H)-hypervariable region cleaved 498 peptides with a highly similar substrate preference (Supplementary Fig. 4a). Employing both peptide libraries is advantageous as each includes within the peptide sequences either the acidic or basic residues, respectively, for full characterization of the substrate specificity. We therefore propose the name "flagellinolysin" and the symbol "FlaMP" for FliA(H) and related metallopeptidase flagellins.

Amino acid occurrences at each substrate subsite were calculated and displayed as heat maps after alignment of the cleaved sequences at the P1↓P1′ scissile bond, where P and P′ residues lie to the N- terminal and C-terminal sides of the scissile bond ↓, respectively (Fig. 4b and Supplementary Fig. 4b). From these, the active site specificity profile from P6 to P6′ was derived (Fig. 4c, Supplementary Fig. 4a for complementary and confirmatory GluC-generated peptide library profiles). Despite the homology with clostridial collagenase, the substrate specificity preference of *C. haemolyticum* flagellinolysin (Fig. 4b, c) was more similar to their mammalian counterparts, the matrix metalloproteinases (MMPs) (Supplementary Discussion), which

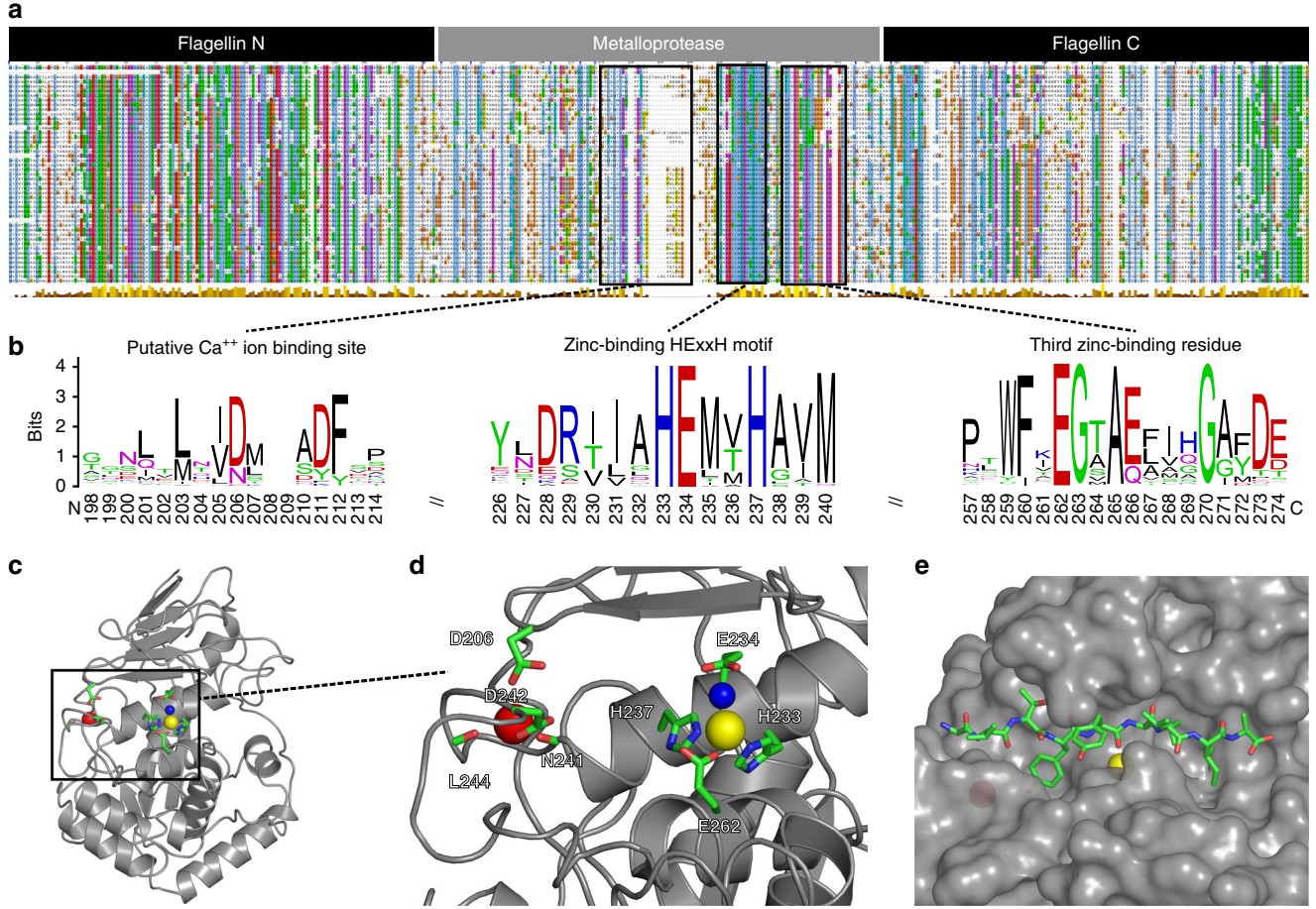

**Fig. 3** Sequence analysis and structural modeling of the proteolytic flagellin family. **a** Multiple alignment of 86 proteolytic flagellin sequences from 74 species, with key functional motifs depicted as sequence logos **b** including a putative Ca$^{++}$ ion-binding site (*left*), the zinc-binding HExxH motif [includes the zinc-binding His233 and His237 and general base Glu234 for hydrolysis] (*middle*) and a motif containing the third zinc-binding residue, Glu262 (*right*). Sequence numbering is according to the FliA(H) sequence, UniProt entry Q8RR94. In addition, the hydrophobic basement-forming residue (Ala265) located below the active site zinc is also conserved. **c** Structural model of proteolytic flagella [FliA(H)-hypervariable region] from *C. haemolyticum* based on the peptidase domain of collagenase H from *C. histolyticum* (PDB 4ar1). **d** The zinc-binding HExxH motif and Ca$^{++}$ ion-binding site with important residues labeled. **e** Peptide-docking model of FliA(H)-hypervariable region complexed with the nonapeptide (AVTYY↓LVIA) showing the scissile bond (↓). The peptide sequence is based on the consensus specificity motif derived from the PICS assay results

cleave a diverse range of proteins in extracellular matrices. Noteworthy, this is in agreement with our structural model (Supplementary Fig. 5). That is: (i) complete absence in the flagellinolysin family of the characteristic "double glycine" motif, which is a key determinant of P1′ substrate specificity in clostridial collagenases[27, 28], and (ii) the presence of a deep pocket S1′ subsite typically found in MMPs.

To further confirm protease activity and substrate specificity, we performed quenched fluorescence (QF) peptide cleavage assays using three peptides: ALG↓L, which closely matches the cleavage specificity determined by PICS; the classical MMP substrate QF24 (PLG↓L); and PLG↓V to test the dependency for leucine in P1′ (Fig. 4d). As demonstrated by the fluorescence emission progress curves, *C. haemolyticum* flagellinolysin readily cleaved both ALG↓L and QF24 (PLG↓L), but did not cleave PLG↓V. This result coincides perfectly with the active site specificity profile determined by PICS and revealed the essential character of the S1′ subsite for bond scission with the absence of PLG↓V cleavage confirming the strong preference for Leu in P1′ similar to MMPs.

Confirming that flagellinolysin was a metalloprotease, we found that peptidase activity was completely abolished in the presence of the metal chelator EDTA (Fig. 4d, red filled circles), indicating divalent cation-dependent proteolytic activity. In contrast, assays with inhibitors of all other protease classes had no effect on activity. Indeed, in our structural model, Asp206 (which substitutes for Glu430 in clostridial collagenase H), forms a Ca$^{++}$ ion-binding site in close proximity to the active site together with the backbone oxygens of Asn241, Leu244, and Asp247 (Fig. 3d). This structural feature was recently identified in clostridial collagenases to stabilize the active site cleft and was indispensable for full proteolytic activity[29]. Notably, we mutated the catalytic residue glutamate 234 to alanine, which resulted in complete loss of cleavage of the ALG↓L quenched fluorescent peptide by the recombinant Glu234Ala flagellinolysin (Supplementary Fig. 6).

To provide a structural context for these active site specificity results, we performed high-resolution peptide docking[30] using the flagellinolysin homology model and a nonapeptide mimicking the consensus specificity motif identified in PICS (AVTYY↓LVIA) (Fig. 3e, Supplementary Fig. 5). Indeed, the peptide proved an excellent fit and further highlighted the deep hydrophobic S2 and S1′ pockets in the active site.

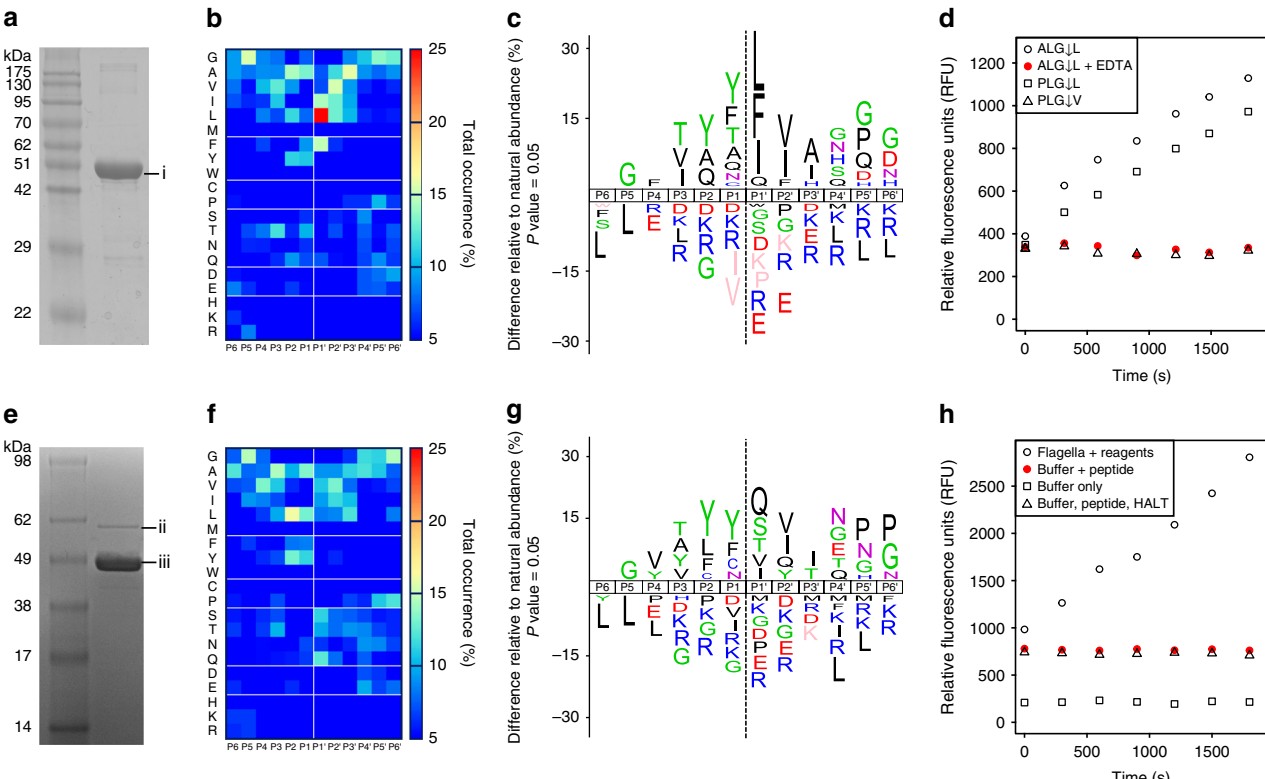

**Fig. 4** Peptidase activity assays of purified recombinant FliA(H)-hypervariable region and flagellar filaments from *C. haemolyticum*. **a** 12% SDS-PAGE of purified FliA(H)-hypervariable region (band "i"), electrophoresing close to that predicted for recombinant thioredoxin-tagged flagellin hypervariable region (50.14 kDa). **b** Amino acid occurrence heat map based on FliA(H)-hypervariable region protease cleaved peptide alignments of 391 cleavage sites identified in a tryptic *E. coli* K12 library. **c** IceLogos of FliA(H)-hypervariable region protease cleaved peptide specificity profiles showing percent differences compared to natural amino acid abundance, with significantly over-represented amino acids shown above the x-axis and under-represented residues below the x-axis. Amino acids that have not been identified are depicted in *pink*. **d** Fluorometric peptidase assays for FliA(H)-hypervariable region protease against three different peptidic substrates (ALG↓L, PLG↓L, PLG↓V) and +/− EDTA as indicated. **e** 10% SDS-PAGE of purified flagellar sheared filaments from *C. haemolyticum*. Band "ii" and "iii" were identified by mass spectrometry as FliA(H) (proteolytic flagellin) and the non-protease-containing structural flagellin (WP_039229459), respectively (Supplementary Fig. 7). **f** Amino acid occurrence heat map and **g** IceLogo based on cleaved peptide alignments of 269 cleavage sites identified in a tryptic *E. coli* K12 library using purified flagellar filaments. **h** Fluorometric peptidase assays for flagellar filaments using the ALG↓L peptide in three assay conditions including EDTA-free HALT protease inhibitors

**Flagellinolysins localize to flagellar filament surfaces**. To confirm that proteolytic flagellins form components of assembled flagellar filaments, we grew *C. haemolyticum* anaerobically, and purified intact flagellar filaments by ultracentrifugation after shearing. Transmission electron microscopy (TEM) imaging showed flagellar filaments to be highly pure and intact (Fig. 5a). SDS-PAGE and liquid chromatography–tandem mass spectrometry (LC–MS/MS) verified that flagellar filaments were composed of two different flagellin proteins: the non-proteolytic "structural" flagellin (NCBI accession WP_039229459), which was the dominant band iii, with 82% protein sequence coverage by LC–MS/MS; and the second most abundant protein band ii, identified as the full-length proteolytic flagellin protein (annotated as FliA(H) in the database), and having a predicted molecular weight of 60.9 kDa (Fig. 4e) with 79% protein coverage by LC-MS/MS, including the HExxH active site catalytic motif (Supplementary Fig. 7).

To visualize the localization of proteolytic flagellins on assembled flagellar filaments, immunogold labeling electron microscopy was performed using affinity purified polyclonal antibodies raised against the proteolytic domain of flagellinolysin (Fig. 5a–d). The specificity of the antibody was verified by immunostaining using both the recombinant protease domain and full-length proteolytic flagellin from purified sheared filaments (Supplementary Fig. 8). Immunogold TEM imaging

revealed highly specific localization of the flagellin protease domain to the surface of individual filaments (Fig. 5b, c), where it was periodically and uniformly distributed, as well as on the surface of flagellar bundles (Fig. 5d).

**C. haemolyticum flagella demonstrate proteolytic activity**. Having verified proteolytic activity of the isolated flagellinolysin peptidase domain and that flagellinolysin was displayed on the exposed surface of flagella, we next confirmed proteolytic activity in the context of whole flagella. To do so, we purified flagellar filaments sheared from cells and analyzed these by both PICS (Fig. 4f, g, Supplementary Fig. 4c) and quenched fluorescent peptide cleavage assays (Fig. 4h). PICS again revealed robust peptidase activity with the purified flagellar filaments cleaving 269 different peptides, and with a substrate specificity matching that observed for the recombinant monomeric protease domain (Fig. 4g). Quenched fluorescence peptide cleavage assays also demonstrated activity of filaments on the peptide ALG↓L, which was abolished in all controls (Fig. 4h). Peptidase activity was significantly reduced in the supernatant following ultra-centrifugation, demonstrating that the observed peptidase activity was filament-associated and not due to free protease monomers in solution. The minor residual (12.8%) activity in the supernatant probably resulted from fragmented filaments that did not

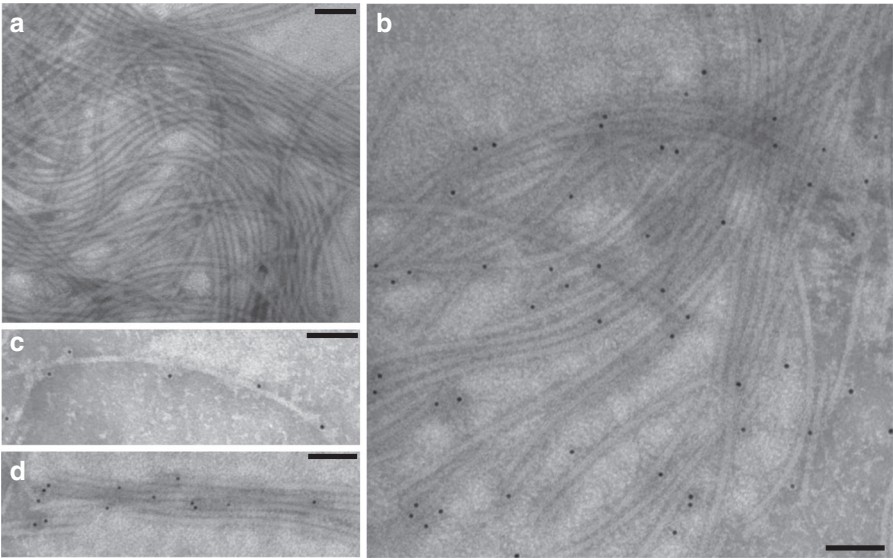

**Fig. 5** Immunogold electron microscopy of intact flagellar filaments from *C. haemolyticum*. **a** Flagellar filaments (control lacking antibody). **b** Flagellar filaments labeled with gold 10-nm nanoparticle-conjugated affinity purified polyclonal antibodies raised against the *C. haemolyticum* flagellin FliA(H)-hypervariable region metallopeptidase domain. Gold nanoparticles can be seen decorating the surface of individual flagellar filaments **c** and flagellar bundles **d**, revealing that the metallopeptidase domains are surface-exposed and distributed evenly throughout filaments. *Scale bars* are 100 nm in length

sediment in these conditions. Finally, we observed minor flagellar filament-associated autocatalytic activity and modest cleavage of bovine native fibronectin, which was also inhibited by EDTA (Supplementary Fig. 9). The loss of the ~440 kDa fibronectin dimer indicates that a C-terminal cleavage occurs, potentially at 2403EY↓LGA2407, which matches the PICS determined cleavage specificity (Fig. 4c). This site lies N-terminal to the C-terminal interchain crosslink sites between Cys2459 and Cys2463 on bovine fibronectin. As plasma fibronectin is a major secretory protein of liver hepatocytes and cellular fibronectin is an important cell adhesive glycoprotein in connective tissues, fibronectin degradation and depolymerization of the multimeric 440-kDa crosslinked form may facilitate *C. haemolyticum* colonization, tissue spreading and necrosis. Taken together, these results demonstrated that *C. haemolyticum* proteolytic flagellins are abundant constituents of assembled flagella, confer peptidase activity to intact filaments, and degrade biological targets of potential physiological relevance to *C. haemolyticum* virulence.

## Discussion

We report the discovery of bacterial flagella with enzymatic activity. Through bioinformatic analysis of the flagellin hypervariable region, we uncovered hidden functional diversity of flagellin-associated domains. These included peptidase M9 and M26[19] enzymatic domains embedded within surface-exposed flagellin hypervariable regions of bacterial flagella. The acquisition of peptidase function may augment its sensory function by generating small peptides and amino acids. Although a remarkable repertoire of domains exists within the rapidly growing collection of bacterial genomes, surprisingly few have been previously characterized. The most common of these, we showed is a catalytically active gluzincin-like metallopeptidase domain present in 74 bacterial species to date, which are scattered across the bacterial tree of life. Despite their abundance in different species, which exceeds that of other characterized flagellin domains such as D2/D3, this proteolytic flagellin family has gone mostly unnoticed to date[31].

Indicative of lateral gene transfer, the proteolytic flagellins are highly dispersed on the bacterial tree of life, even at the inter-phylum level. Interestingly, we observed evidence for two separate mechanisms of lateral transfer: one involving possible transfer of a full-length proteolytic flagellin gene (e.g., flagellin gene insertion into *Comamonas aquatica*, Supplementary Fig. 1); and a second involving a partial insertion of only the protease domain into the flagellin hypervariable region (Supplementary Fig. 2). Consistent with the latter mechanism, *C. haemolyticum* and several closely related genes possess the metallopeptidase domain, but it is completely absent in the larger flagellin family of clostridial flagellins from which *C. haemolyticum* flagellin is derived (Supplementary Fig. 2). This further implies an ancestral protease insertion event in the ancestor of the *C. haemolyticum* flagellin subfamily. Thus, both full and partial gene insertion mechanisms have been utilized.

As flagellinolysin domains form a discrete family that is distant from M9, as well as other gluzincin families, e.g., M2 and M4[19], the exact identity of the ancestral protease insertion is undetermined. If the proteolytic flagellin hypervariable region evolved by acquisition of a collagenase-related domain, a subsequent shift in substrate specificity must have occurred, in part through substitution of the key glycine–glycine motif, perhaps driven by selection for saprophytic degradation of environmental or biofilm peptides rather than collagen. Indeed, we were unable to demonstrate native or denatured type I collagen cleavage by flagellinolysin that lacks collagen binding exosite domains essential for collagen triple helicase activity[28, 32].

To test our bioinformatic prediction of flagellin-associated proteolytic function, we focussed on *C. haemolyticum*, an important animal pathogen and cause of bacillary hemoglobinuria[33]. In this organism, proteolytic flagellins assembled as abundant components of flagellar filaments yielding extracellular peptidase activity. Considering a relative abundance of up to 10% in assembled filaments, proteolytically active flagella would consist of 1000–2000 active sites per filament. Thereby, proteolytic flagellins form the largest protease assemblies known to date with up to ~ 20,000 flagellin copies co-polymerized in a single macromolecular structure ~ 10 μm long. Especially when considering the extra cell surface area provided by flagellar filaments, the immobilization of enzymes on flagella results in highly localized and amplified enzymatic gradients in controlled environments

around a single cell. When directed by cell receptors, flagellated bacteria turn and undergo directed motion toward ascending gradients nutrients and amino acids, which are potentially generated by the flagella itself in cis, to focally improve the nutrient environment for increased pathogenic potential.

Previous studies have exploited the functional potential of the flagellin hypervariable region in vitro by replacing it with adhesive peptides[34, 35], fluorescence domains[36], and glycosyl hydrolases[37]. However, prior to our discovery, only structural and adhesive functions for flagellins have been observed in nature. Hence, the newly discovered metallopeptidase flagellinolysin family (FlaMP) we report here highlights an enzymatic role for bacterial flagellar filaments. It also raises the possibility of enzymatic or other unknown functionality in existing flagellin hypervariable region domains of unknown function, e.g., the D3 domain found in *Salmonella* flagellins. Future work to explore proteolytic flagellins in additional species will examine the numerous potential biological pathways impacted by flagellum-mediated extracellular proteolysis, including biofilm or host-tissue adherence/colonization, virulence, bacterial defense, or extracellular cell–cell communication.

Lastly, our findings provide a fascinating and multilayered story of molecular evolution, involving not only protein domain recombination, but also lateral gene transfer. First, a collagenase-related gene appears to have inserted into a flagellin hypervariable region, presumably within a collagenase-containing lineage such as *Clostridium*. This is consistent with numerous studies that have documented intragenic recombination in flagellins, which serves as a mechanism for antigenic diversification[38–43]. It is reasonable to assume that this domain insertion likely happened after the evolution of microbial collagenase and MMP-like proteins. The proteolytic flagellin gene then spread to other microorganisms through lateral transfer (Fig. 2, Supplementary Fig. 1), possibly within host-associated and aquatic microbiomes. The remarkable degree of lateral transfer combined with the observed enzymatic activity and clear evidence of selection at the sequence level, suggest that these transfers confer a functional benefit to their recipients.

## Methods

**Computational methods**. The CDART tool[44] was used to identify all proteins in the NCBI non-redundant protein database containing a predicted flagellin N-hypervariable region-C domain architecture (search performed on 31 January 2017). Identified sequences possessing a central putative gluzincin domain within the flagellin hypervariable region were retrieved and aligned using MUSCLE version v3.8.31[45] with default parameters, and visualized in Jalview version 2.80b1[46]. A structural model of the FliA(H)-hypervariable region was generated using the modeling servers SWISS-MODEL[47] and iTASSER[22] with the crystal structures of clostridial collagenase G (PDB entry 2y3u), H (PDB entry 4ar1), and T (PDB entry 4ar8) as templates[28, 29]. Phyre 2.0's intensive modeling procedure was also used and yielded highly similar results[21]. Energy minimization and secondary structure assignment were performed within the UCSF Chimera package[48]. Peptide docking was initiated with models based on available collagenase complex structures[28, 29] and the PICS peptide cleavage specificity results using the Rosetta FlexPepDock web server[30], allowing full peptide flexibility and side-chain flexibility to the protease. Molecular graphic figures were made using the molecular visualization system PyMOL (www.pymol.org). Phylogenetic analysis was performed by mapping proteolytic flagellin presence/absence on to a recently constructed tree of life[49].

**Protein expression and purification**. FliA(H)-hypervariable region (Lys[149]–Ile[448], NCBI RefSeq protein sequence accession number WP_039229452.1) cDNA was constructed by gene synthesis and subcloned into the *KpnI* and *XhoI* sites in pET-32a, and expressed in *Escherichia coli* BL21(DE3) cells (Bio Basic Inc.). Protein expression was induced using 0.1 mM IPTG and continued for 3 h at 30 °C. After induction, cells were collected by centrifugation (4000 r.p.m., 10 min). The pellet was re-suspended in 50 mM Tris, 300 mM NaCl, pH 8.0. The cell suspension was sonicated for 30 min in ice bath (sonication power 450 W, a sonication cycle: working time 2 s and pause time 6 s). Additional sonication cycles were repeated until the solution became translucent. The cell debris was removed by centrifugation at 12,000 r.p.m. for 20 min. After ultrasonication, supernatant was purified using HisTrap column (Ni$^{2+}$ separose column, GE healthcare). The sample was applied to the column and washed with at least five column volumes of the washing buffer (50 mM Tris, 300 mM NaCl, pH 8.0). The protein was subsequently eluted with 20 mM Tris-HCl (pH 8.0), 300 mM imidazole, 300 mM NaCl. The protein was then applied to a Superdex 75 column (GE Healthcare) equilibrated in 1X PBS, pH 7.0 while monitoring OD at 280 nm, and the eluted fractions were analyzed via SDS-PAGE. Typically, 2.5 mg of highly pure protein were obtained per 500 ml of *E. coli* culture. Uncropped gel images used in the preparation of figures are shown in Supplementary Fig. 10.

**Proteomic identification of protease cleavage sites assay**. PICS assays using whole proteome peptide libraries were performed as described in detail previously[25, 26, 50–52]. To produce *E. coli* proteome-derived peptide libraries, cell pellets collected from *E. coli* K12 JM109 cultures (New England Biolabs) were lysed in the presence of protease inhibitors and cell debris was removed by centrifugation. A concentration of 4 M guanidine hydrochloride was used to denature soluble proteins, and cysteine side chains were alkylated using iodoacetamide. After chloroform/methanol precipitation, the pelleted protein interphase was re-suspended, and trypsin (TPCK treated, Sigma-Aldrich) digestion was applied at a protease to proteome ratio of 1:100 (w/w). Reductive dimethylation was used to block primary amines and samples were desalted using size exclusion chromatography and further purified using reversed-phase chromatography. Peptide eluates were concentrated, re-suspended in water, and stored as 200 µg peptide libraries at –80 °C. PICS cleavage assays were performed by incubating 200 µg of peptide library with recombinant FliA(H)-HVR protein. Cleaved peptides were selectively biotinylated, affinity purified and desalted using reversed-phase solid phase extraction. Eluates were vacuum dried to near dryness and stored at –80 °C until LC–MS/MS analysis on a high-resolution quadrupole Time-Of-Flight mass spectrometer (Impact II, Bruker Daltonics), as detailed below.

**Spectrum to peptide matching and identification**. Peptides were identified at a ≤ 1% false discovery rate from the UniProt *E. coli* K12 database (November 2013) using two search engines, Mascot v2.4.1 (Matrix Science, London, UK) and X! Tandem[53], in conjunction with PeptideProphet[54] as implemented in the Trans-Proteomic Pipeline v.4.6. Search parameters included a mass tolerance of 15 p.p.m. for the parental ions and 0.1 Da for fragment ions, and allowed up to two missed cleavages. The following fixed peptide modifications were used: carbamidomethylation of cysteine residues (+ 57.02 Da) and dimethylation of lysine ε-amines (+ 28.03 Da). N-terminal dimethylation of uncleaved library carryover peptides (+ 28.03 Da), methionine oxidation (+ 15.99 Da), and thioacylation of protease-generated neo N-termini (+ 88.00 Da) were set as variable modifications. Peptide lists from both search engines were combined within the Trans-Proteomic Pipeline for further analysis. A web-based bioinformatics tool, WebPICS[55] was used to reconstruct the non-prime side of each identified unique cleavage site[28, 29]. Obtained cleavage sites were aligned along the scissile peptide bond and visualized as heat maps in GnuPlot (www.gnuplot.info) and iceLogos[56]. The MS raw data associated with the present paper are available upon request.

**Quenched fluorescence protease activity assays**. Stock solutions (1.0 mM) of synthetic quenched fluorescence (QF) peptide substrates (GenScript Inc.) were dissolved in DMSO and working stocks (100 µM) were prepared using the molar extinction coefficient of the conjugated quencher, (2,4)-dinitrophenyl, of 6.985 cm$^{-1}$ mM$^{-1}$ at 400 nm. Purified recombinant FliA(H)-hypervariable region protease was incubated at a final concentration of 0.5 µM with 10 µM QF-peptide substrate in 100 µl of 150 mM NaCl, 10 mM CaCl$_2$, 50 mM HEPES, pH 7.5, in the presence of HALT protease inhibitor cocktail (Life Technologies) plus/minus 20 mM EDTA at 37 °C. Three different quenched fluorescent peptides (PLG↓L, Mca-Pro-Leu-Gly-Leu-Dpa-Ala-Arg; ALG↓L, Mca-Ala-Leu-Gly-Leu-Dpa-Ala-Arg; and PLG↓V, Mca-Pro-Leu-Gly-Val-Dpa-Ala-Arg)[51] were tested using a multi-wavelength fluorescence scanner (POLARstar OPTIMA, BMG Lab technologies). The excitation and emission wavelengths were 320 and 405 nm, respectively. Fluorescence was measured in arbitrary units over 30 min at 45 s intervals. Values are means of triplicate measurements.

**Purification and analysis of flagellar filaments**. Cultures of *C. haemolyticum* were grown anaerobically (in an atmosphere of 80% N$_2$, 10% CO$_2$ and 10% H$_2$) overnight at 30 °C and collected by centrifugation. The cell pellet was re-suspended in 0.1 M Tris-HCl, pH 7.0 in 1/20th of the culture original volume. Flagella were removed from cells by mechanical shearing using a Teflon tissue homogenizer. The flagellar filaments were then purified after removing whole cells by two centrifugation cycles at 5000×*g*, 15 min each, followed by ultracentrifugation at 130,000×*g* for 1 h (70 Ti rotor, Beckman Coulter Canada Inc.). The pellets containing the flagella were then re-suspended in ultrapure water followed by a second round of ultracentrifugation (130,000×*g*, 1 h). The purified flagellin was quantified using BCA protein assay and re-suspended in 50% glycerol and stored at –20 °C[57]. Fidelity of purification was confirmed by SDS-PAGE (12%). Protein bands of 62 kDa (proteolytic flagellin) and 46 kDa (structural flagellin) were excised and analyzed by in-gel trypsin digestion and sequenced by LC–MS/MS (Advanced Analysis Centre, University of Guelph, Guelph, Ontario, Canada).

**Immunogold transmission electron microscopy**. The concentration of affinity purified primary rabbit anti-FliA(H) antibody against isolated flagellar proteins used for immunogold TEM was optimized and guided by enhanced chemiluminescent (ECL)-western blotting of the purified recombinant FliA(H) protein, expressed with a thioredoxin tag (Trx), and isolated flagellar proteins. The secondary antibody used was goat HRP-conjugated anti-rabbit IgG.

Isolated flagella (50 ng) from *C. haemolyticum* were directly pipetted on formvar-coated nickel grids (Ted Pella Inc. Lot# 221014; 200 mesh, Ni; Prod No. 01800 N) and air-dried for 15 min. The grids were blocked in 100 μl of blocking buffer (1X PBS + 0.1% Triton X100 + 1% BSA) for 2 h before incubation with 100 μl of anti-FliA(H) primary antibody diluted 1:50 from stock concentration of 2 mg ml$^{-1}$ in wash buffer (1X PBS + 1% BSA). The primary antibody incubation was performed overnight in a wet chamber at 4 °C after which the grid was washed three times with wash buffer. For immunogold labeling, grids were incubated in goat anti-rabbit IgG-Gold (10 nm) secondary antibody, diluted 1:100 in the wash buffer, for 1 h. The grids were negatively stained with 1% (w/v) ammonium molybdate. TEM was performed using a Philips CM10 microscope at the Electron and Confocal Microscopy Facility, Department of Biology, University of Waterloo.

**Statistical methods**. The statistical approaches used in PICS are standardized and involve calculation of position-specific amino frequencies in an experimental *versus* reference set of peptides, and visualization as sequence logos[25, 56] (e.g., Fig. 4). Residue frequencies were calculated based on iterative sampling of peptides from the experimental and reference sets, and *P*-values were estimated using standard *t*-tests (distributions are inferred to be normal based on central limit theorem). Use of a large cohort (hundreds to thousands) of individual cleaved sequences provides robust statistical confidence for determination of subsite specificity[25].

Statistical confidence for proteins and peptides identified by LC–MS/MS was quantified with PeptideProphet[54] using false discovery rate calculations, which estimate the proportion of total predictions that are false due to chance.

Homologs of proteolytic flagellins were identified using NCBI's CDART tool and PSI-BLAST[20]. *C. haemolyticum* flagellinolysin (UniProt entry Q8RR94) was used as an initial query in two rounds of PSI-BLAST, and an *E*-value threshold of 0.001 to narrow high-confidence matches.

**Mass spectrometry**. LC–MS/MS analysis was performed on a nano-LC system (EASY-nLC 1000, Thermo Scientific, USA) coupled to a high-resolution quadrupole Time-Of-Flight mass spectrometer (Impact II, Bruker Daltonics) using the CaptiveSpray ionization source (Bruker Daltonics), a 2-cm long, 75-μm inner diameter fused silica trap column, and a 20 cm long, 50-μm inner diameter fused silica fritted analytical column. The trap column was packed with 5-μm diameter Aqua C18 beads (Phenomenex, USA), whereas the analytical column was packed with 1.9-μm diameter Reprosil-Pur C18-AQ beads (Dr. Maisch, Ammerbuch, Germany). Peptides were eluted using a 0 – 80% gradient of organic phase over 90 min. Buffer A was 0.1% formic acid and buffer B was 100% (v/v) acetonitrile with 0.1% formic acid. MS/MS data were acquired automatically using the otof-Control software (Bruker Daltonics) for information-dependent acquisition. Error of mass measurement was usually within 5 p.p.m. and was not allowed to exceed 10 p.p.m. Peak lists and mzXML-files of the acquired high resolution Time-Of-Flight mass spectrometer data were created using the Compass DataAnalysis software 4.2 (Bruker Daltonics).

**LC–MS/MS of recombinant FliA(H)-hypervariable region**. Tandem mass spectrometry was performed essentially as described in detail by Gundry et al.[58]. In brief, free sulfhydryl groups of 10 μg of FliA(H)-hypervariable region protease in 1X phosphate buffer saline (PBS; 10 mM Na$_2$HPO$_4$, 1.8 mM KH$_2$PO$_4$, 137 mM NaCl, 2.7 mM KCl, pH 7.4) were alkylated with iodoacetamide (20 mM iodoacetamide, 1 h, 20 °C) after initial cysteine reduction with 10 mM dithiothreitol (DTT; 30 min, 20 °C). Adding further DTT (10 mM, 15 min, 20 °C) ended labeling. Trypsin Gold (Promega) was added in a 1:50 w/w ratio and the sample was digested overnight at 37 °C. The reaction was stopped by adding formic acid to a final concentration of 2.5% and desalted by C18-STAGE tip[59] prior mass spectrometry analysis.

LC–MS/MS analysis was performed as described below. Peptides were identified at a 1% false discovery rate from a combined UniProt database, containing both human and *E. coli* proteins and supplemented by our target protein, thioredoxin-tagged FliA(H)-hypervariable region (flagellinolysin). Two search engines, Mascot v2.4.1 (Matrix Science, London, UK) and X! Tandem[53] were used in conjunction with PeptideProphet[54] as implemented in the Trans-Proteomic Pipeline v.4.6. Search parameters included a mass tolerance of 15 p.p.m. for the parental ions and 0.1 Da for fragment ions, and allowed for one missed cleavage. Carbamidomethylation of cysteine residues (+ 57.02 Da) was used as fixed modification, whereas methionine oxidation (+ 15.99 Da) was set as a variable. Peptide lists from both search engines were combined within the Trans-Proteomic Pipeline for further analysis. The MS raw data associated with the present paper are available upon request.

**In-gel digestion and mass spectrometry**. In-gel digests were performed as described by Huesgen et al.[60]. Recombinant flagellinolysin was resolved by 12%

SDS-PAGE, stained with Coomassie G-250, and nine bands were excised, destained with 60% acetonitrile, 20 mM ammonium bicarbonate, and then washed in 100% acetonitrile before lyophilization. Gel bands were rehydrated with 10 μl Trypsin Gold (Promega) in 20 mM ammonium bicarbonate (12 ng/μl) by passive diffusion for 1 h at 4 °C. Excess solution was removed and 20 μl 20 mM ammonium bicarbonate was added to the gel plugs and digested overnight at 37 °C. Supernatants were removed and three rounds of active extraction were performed using 20 μl of 1% formic acid, and two times 20 μl of 30% acetonitrile, 1% formic acid. All supernatants were pooled and concentrated to 10 μl using a SpeedVac, and 1–2 μl were analyzed by LC–MS/MS using the same set-up as above, but 30 min gradients. Search parameters included 15 p.p.m. tolerance for MS and 0.10 Da for MS/MS, variable methionine oxidation and propionamide cysteine, and a maximum of one missed cleavage.

**Data availability**. The authors declare that all data supporting the findings of this study are available from the corresponding authors upon reasonable request.

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

## Acknowledgements

This work was supported by the Natural Sciences and Engineering Research Council of Canada, through a Discovery Grant to A.C.D., a postdoctoral fellowship from the Michael Smith Foundation of Health Research (MSFHR) to U.E., a Canada Research Chair in Metalloprotease Proteomics and Systems Biology (C.M.O.), and project grants from the Canadian Institutes of Health Research (CIHR), as well as with infrastructure grants from the MSFHR and the Canada Foundation for Innovation to C.M.O. We would like to thank Glen Armstrong, Klaus Winzer, and Brendan McConkey for helpful discussions. We also thank Simon Chuong and Mishi Groh for technical assistance on immunogold electron microscopy. We gratefully acknowledge Nestor Solis and Theo Klein for invaluable assistance with the mass spectrometer.

## Author contributions

U.E., H.B., G.M., J.C., I.W. and J.A. performed experiments. U.E., M.J.M., T.H., T.C.C., J.A., C.M.O. and A.C.D. performed data analysis and interpretation. U.E., H.B., M.J.M., C.M.O. and A.C.D. wrote and prepared the manuscript.

## Additional information

**Competing interests:** The authors declare no competing financial interests.

