## [Peer Review file · Nature Communications]

Reviewers' comments:

Reviewer #1 (Remarks to the Author):

The authors have described a novel metallopeptidase domain found within the variable region of the flagellin protein of bacteria. The methodology is sound, and the flagellinolysin from *Clostridium haemolyticum* has been extensively characterized with the use of synthetic substrates, inhibitors, and peptides derived from the *E. coli* proteome analysed by PICS. I have only minor points.

Minor points

1. The metallopeptidase is identified as a Glu-zincin because a well conserved Glu is present following the HEXXH zinc-binding motif and it is correctly assumed that the Glu provides the third metal ligand with the two histidines from the HEXXH motif. Given that the flagellin from *Clostridium haemolyticum* has been expressed, it would have been possible to prove this point by site-directed mutagenesis. The authors speculate that the flagellinolysin metallopeptidase domain is most closely related to the M9 family of metallopeptidases, but there is no evidence for this: the M9 family has been chosen probably because the best characterized member is from the same organism. I see no reason why a thermolysin homologue (family M4) could not be the ancestor of the flagellin peptidase domain. The flagellinolysin domains form a discrete family which is equally distant to M4, M9 and the other Glu-zincin families (some of which, such as M2 - angiotensin-converting enzyme - are eukaryotic).

2. In the discussion, metallopeptidase family M26 is introduced for the first time (it also occurs in Table 1). This is another Glu-zincin family, and perhaps a sentence should be added to the Introduction to explain the various Glu-zincin families and how they differ.

3. The description of the potential horizontal gene transfer (HTG) could be expanded. The unusual distribution of this domain, namely that it is present in the flagellin of one species but not in the flagellin of a closely related species, is intriguing: but it is probably more complicated than a simple HTG. It would seem that either a) a gene encoding a metallopeptidase has been inserted in the flagellin gene in one species, and because this is advantageous a horizontal gene transfer replaces the old flagellin gene without the peptidase-encoding region for one that does include it; or b) the variable region of the flagellin gene is like a cassette and a region encoding a domain can be shuffled in or out.

4. Page, 10, lines 29: WP_039229452 is a RefSeq identifier for a protein sequence derived from the NVBI Protein database; this should be explained.

5. Page 12, lines 13-15: I presume the downward pointing arrow is the site of cleavage in these synthetic substrates. This should be explained.

Reviewer #2 (Remarks to the Author):

The authors use bioinformatics approaches to show that some flagellin genes encode enzymes between the conserved N and C terminal polymerization domains of flagellin. They purify just the metalloprotease domain from one of these genes from *Clostridium haemolyticum*, show it had enzymatic activity, and characterize the target cleavage site. They further show by TEM immunogold that the metalloprotease containing flagellin was in fact incorporated into flagella as a minority component and that assembled and purified flagellar filaments also have protease activity. This is a very interesting piece of work and calls attention to a large class of flagellin proteins that contain enzymatic domains (I had never heard of this before). The case study they've chosen is likely important to the pathogenesis of the organism and their data suggest that the poorly understood D3 domains of unknown function, such as those found in *E. coli* and *S. enterica* might also have additional functions separate from motility. The only experiment I had wished was

included was a protease assay using full length *C. haemolyticum* flagellin protein transgenically expressed and purified from *E. coli* (similar to Fig 4A,D). If protease domain is functional but the full length protein is not, it may suggest that the flagellin must be incorporated into the filament to achieve activity and thus avoid the complication of an active protease expressed in the cytoplasm prior to export and polymerization. A great paper sure to change the way flagellar biologists and pathogen researchers look at flagellin.

Page 3 line 20. Consider also including the abbreviation (PAMP) as it might be more recognizable than the full descriptor.

Page 4 line 11. Clarify "natural". Do the authors mean "commonly-studied"?

Page 6 line 15. What is PICS?

Page 7 line 12. Clarify the point that the organism encoded two different flagellin proteins. Perhaps make this idea more explicit.

Page 7 line 29. Clarify the origin of the flagellar filaments. Were these filaments sheared from cells and purified?

Page 8, line 20. Omit "extremely". The paper is strong and emphatics are unnecessary.

Page 9 line 15-17. Omit.

Page 9 line 29. Omit "extreme". Again the emphatic is unnecessary and it is difficult to qualify the extremeness of horizontal gene transfer.

End of discussion. The authors might consider mentioning that their results suggest the possibility that flagellin D3 domains of unknown function, such as that found in *E. coli* and *S. enterica* might constitute as yet-unknown enzymatic activities as well.

Reviewer #3 (Remarks to the Author):

What are the major claims of the paper?

The authors report the first case of enzymatic function associated with bacterial flagella.

Are they novel and will they be of interest to others in the community and the wider field? Do you feel that the paper will influence thinking in the field?

The authors claim is novel and in my opinion, will be of interest to the scientific community. The primary role of the flagellum is locomotion, but it can also function as a sensory organelle. In addition to these roles, the authors' analysis suggests that flagellum acquired a metallopeptidase gene, which then spread by lateral gene transfer. The peptidase is inserted in the hypervariable region, and it should be facing the exterior of the flagellar filament.

The peptidase acquired by the flagellum carries sequence motifs found in the structure of catalytic domains of bacterial collagenase that Eckhard had previously determined. Evolutional pressure appears to have retained the two features found in the collagenase, i.e. the Ca²⁺ sensors and the catalytic machinery to cleave the peptide bond. However, truncated enzymes lost structural features necessary to recognize and potentially unwind the triple helix of collagen. The peptidase domain was cloned and expressed by the authors and analyzed for its substrate specificity. PICS MS/MS assays of peptides generated by the peptidase do not suggest that it is a collagenase, but it appears to selectively cleave between Tyr and Leu in the AVTY^Y↓LVIA sequence.

The EM clearly shows that the peptidase is located in a regular interval in the surface of intact flagellar filaments isolated from *C. haemolyticum*. The isolated flagellar filaments exhibited peptidase activity. The PICS MS/MS assays of the isolated filaments indicated Leu was replaced with Gln (GV~~T~~YY↓QVIN), possibly because of minor structural changes induced by the filamentous arrangement.

The authors also catalogued 85 flagellins with peptidase domains inserted at the hypervariable region. A functional benefit for bacteria possessing peptidase activity in flagellins is unknown at this time. The acquisition of peptidase function may augment its sensory function by generating small peptides and amino acids. Since the peptidase characterized was demonstrated to be highly specific for a certain peptide sequence, it seems likely that it has a specific purpose. Perhaps the motor-driven movement of the flagella combined with proteolytic activity allows the cell to "drill" a hole into a protein-rich matrix during cell invasion.

It reminds me of the uremia of southern India. Because of its danger to the user, it is taught last. Likewise, proteolytic flagella appears to have been "taught" to select bacteria.

Reviewer #4 (Remarks to the Author):

The paper by Overall, Doxey and colleagues reports on the identification of a new flagellin family exhibiting proteolytic activity. Lateral gene transfer led to insertion of a zinc-dependent metalloprotease domain into the hypervariable region of flagellin. Bioinformatics revealed this a widespread event between diverse bacterial phyla, including Firmicutes, Beta- and Gammaproteobacteria. Interestingly, some of these protease domain containing flagellins, named flagellinolysins, have additionally been identified in animal pathogens, e.g. in *Clostridium haemolyticum*. This gives rise to potential functions of flagellinolysins as pathogenic factors. Other possible functions include bacterial defense mechanisms or cell-cell communication.

For biochemical and functional characterization the authors produced recombinant flagellinolysin from *C. haemolyticum*, called FliA(H), and investigated its catalytic activity employing a proteomics approach for the identification cleavage site specificity (PICS). Interestingly, the specificity turned out to be MMP-like, with preference for leucine in P1', which was additionally validated in single peptide cleavage assays. As control EDTA as metalloprotease inhibitor was used and a catalytically inactive mutant of FliA(H) was produced. For more physiological assays, the authors isolated intact flagellar filaments from anaerobically cultured *C. haemolyticum* and again proofed catalytic activity employing PICS validating the results observed with recombinant enzyme.

The discovery of proteolytically active flagellar filaments is a striking finding important for a wide multidisciplinary field. It provides a new fascinating research aspect with possible impact on the understanding of host-microbiome interaction. All experiments and data presented herein are of high quality and support the discovery of flagellinolysins as active proteases.

However, one additional experimental approach should be considered prior publication. As nicely stated by the authors "With ~20,000 flagellin copies per ~10 μ m flagella this assembles the largest proteolytic complex known to date." and with regard to such high abundance of enzymatic activity on *C. haemolyticum*, I suggest to analyze potential substrates in the host proteome. *C. haemolyticum* can infect man and animals, in some cases through the intestinal epithelium. The group of Dr. Overall developed a very effective technique for the identification of protease substrates. The N terminomics approach TAILS will provide evidence, if the proteolytic activity of FliA(H) specifically cleaves cell surface or secreted proteins produced by cultured intestinal cells. This would further support the importance of this kind of protease and would strengthen the biological part of this study.

Reviewer #1:

The authors have described a novel metallopeptidase domain found within the variable region of the flagellin protein of bacteria. The methodology is sound, and the flagellinolysin from Clostridium haemolyticum has been extensively characterized with the use of synthetic substrates, inhibitors, and peptides derived from the E. coli proteome analysed by PICS. I have only minor points.

Minor points

1. The metallopeptidase is identified as a Glu-zincin because a well conserved Glu is present following the HEXXH zinc-binding motif and it is correctly assumed that the Glu provides the third metal ligand with the two histidines from the HEXXH motif. Given that the flagellin from Clostridium haemolyticum has been expressed, it would have been possible to prove this point by site-directed mutagenesis.

Done as suggested. The Glu 234 in the HEXXH motif was mutated to Ala. Fluorometric peptidase assays for FliA(H)-hypervariable domain (wild type) *versus* catalytic inactive mutant FliA(H)-hypervariable domain (E234A) using the quenched fluorescent peptidic substrate with the sequence ALGL showed complete loss of activity (**Supplementary Fig. 6**). Assays were performed over three time points, the results normalized and calculated from total of six data points per sample (P7, L8-11). Extensive site directed mutation analyses of the active site, including the three Zn⁺⁺ ion binding residues, and other characteristic residues in many metalloproteinases by the protease community have repeatedly confirmed these residues essential involvement in catalytic function, yet for the Zn⁺⁺ ion ligators, also shown structural destabilization that often can not be dissociated from loss of activity upon their mutation. For this reason we do not consider further mutation analysis insightful.

We have further addressed the Reviewer's point by softening the labeling of the Gluzincin domain to "putative" or "Gluzincin-like" instead of implying it as a definite gluzincin family domain. See P 5, L 8,20; P 11, L 16; and Table 1.

The authors speculate that the flagellinolysin metallopeptidase domain is most closely related to the M9 family of metallopeptidases, but there is no evidence for this: the M9 family has been chosen probably because the best characterized member is from the same organism. I see no reason why a thermolysin homologue (family M4) could not be the ancestor of the flagellin peptidase domain. The flagellinolysin domains form a discrete family which is equally distant to M4, M9 and the other Glu-zincin families (some of which, such as M2 - angiotensin-converting enzyme - are eukaryotic).

Corrected as suggested. Fair point. Flagellinolysins do indeed form a discrete sequence family and we cannot rule out other gluzincin domain families as being more closely related. We have included the following sentence in the manuscript to address this:

Pg 9, Ln 26: "As flagellinolysin domains form a discrete family that is distant from M9 as well as other

gluzincin families, e.g., M2 and M4¹⁹, the exact identity of the ancestral protease insertion is undetermined.”

2. In the discussion, metallopeptidase family M26 is introduced for the first time (it also occurs in Table 1). This is another Glu-zincin family, and perhaps a sentence should be added to the Introduction to explain the various Glu-zincin families and how they differ.

Done as suggested. We now describe M26 when it is first encountered (P 4, L 15-17): “Sequences annotated as Peptidase M26¹⁹ family domains, which include metallopeptidases that cleave mammalian IgA, were also identified within some flagellins (Table 1).”

3. The description of the potential horizontal gene transfer (HTG) could be expanded. The unusual distribution of this domain, namely that it is present in the flagellin of one species but not in the flagellin of a closely related species, is intriguing; but it is probably more complicated than a simple HTG. It would seem that either a) a gene encoding a metallopeptidase has been inserted in the flagellin gene in one species, and because this is advantageous a horizontal gene transfer replaces the old flagellin gene without the peptidase-encoding region for one that does include it; or b) the variable region of the flagellin gene is like a cassette and a region encoding a domain can be shuffled in or out.

Done as suggested. This is an insightful comment, and we agree with the reviewer. We do in fact see evidence for both of the reviewer’s suggested mechanisms: 1) lateral transfer of full-length flagellins (Supplementary Fig. 1, see *Comamonas aquatica* example), which we now point out clearly in the text; and 2) shuffling of the proteolytic hypervariable region into existing flagellins. As evidence of the latter mechanism, we have included new analyses: A phylogenetic tree of the most closely related flagellins to the protease-containing flagellin *C. haemolyticum* flagellinolysin based on an alignment of only the flagellin N-terminal domain (see **new Supplementary Fig. 2**). Only the immediate relatives are included as there are tens of thousands of more distantly related flagellins. Interestingly, the top 5 most closely related sequences to *C. haemolyticum* possess the protease domain insertion whereas it is completely absent from the larger flagellin family from which *C. haemolyticum* flagellin is derived. This implies an ancestral protease insertion event in the ancestor of the *C. haemolyticum* flagellin subfamily. If this were not the case and only mechanism 1 was at work, all of the proteolytic flagellins should cluster as a group based on an N-terminal domain alignment.

We have added this information to the text as shown below:

P 5, L 18-20: “FliA(H) forms a subfamily of flagellins containing a metallopeptidase domain within the hypervariable region, and the absence of this domain in related sequences implies an ancestral insertion event (Supplementary Fig. 2). ”

P 9, Second Paragraph: “Indicative of lateral gene transfer, the proteolytic flagellins are highly dispersed on the bacterial tree of life, even at the inter-phylum level. Interestingly, we observed evidence for two separate mechanisms of lateral transfer: one involving possible transfer of a full-length flagellin gene (e.g., flagellin gene insertion into *Comamonas aquatica*, Supplementary Fig. 1); and a second involving a partial insertion of only the protease domain into the flagellin hypervariable region (Supplementary Fig. 2). Consistent with the latter mechanism, *C. haemolyticum* and several closely related genes possess the metallopeptidase domain, but it is completely absent in the larger flagellin family of clostridial flagellins from which *C. haemolyticum* flagellin is derived (Supplementary Fig. 2). This further implies an ancestral protease insertion event in the ancestor of the *C. haemolyticum* flagellin subfamily. Thus, both full and partial gene insertion mechanisms have been utilized.”

4. Page, 10, lines 29: WP_039229452 is a RefSeq identifier for a protein sequence derived from the NVBI Protein database; this should be explained.

Done as suggested. We have added (now P 11, L 30-31): "FliA(H)-hypervariable region (Lys¹⁴⁹–Ile⁴⁴⁸, NCBI RefSeq protein sequence accession number WP_039229452.1) ...".

5. Page 12, lines 13-15: I presume the downward pointing arrow is the site of cleavage in these synthetic substrates. This should be explained.

Done as suggested. The Reviewer is correct. This is a standard designation in the protease community and we have now defined this to be more accessible to the multidisciplinary community: P6, L10-13: "Amino acid occurrences at each substrate subsite were calculated and displayed as heat maps after alignment of the cleaved sequences at the P1↓P1' scissile bond, where P and P' residues lie to the N- and C-terminal sides of the scissile bond ↓, respectively (Fig. 4b and Supplementary Fig. 4b)." In the Figure 3e legend we now also state: "(e) Peptide-docking model of FliA(H)-hypervariable region complexed with the nonapeptide (AVTYY↓LVIA) showing the scissile bond (↓)."

Reviewer #2:

The authors use bioinformatics approaches to show that some flagellin genes encode enzymes between the conserved N and C terminal polymerization domains of flagellin. They purify just the metalloprotease domain from one of these genes from Clostridium haemolyticum, show it had enzymatic activity, and characterize the target cleavage site. They further show by TEM immunogold that the metalloprotease containing flagellin was in fact incorporated into flagella as a minority component and that assembled and purified flagellar filaments also have protease activity. This is a very interesting piece of work and calls attention to a large class of flagellin proteins that contain enzymatic domains (I had never heard of this before). The case study they've chosen is likely important to the pathogenesis of the organism and their data suggest that the poorly understood D3 domains of unknown function, such as those found in E. coli and S. enterica might also have additional functions separate from motility. The only experiment I had wished was included was a protease assay using full length C. haemolyticum flagellin protein transgenically expressed and purified from E. coli (similar to Fig 4A,D). If protease domain is functional but the full-length protein is not, it may suggest that the flagellin must be incorporated into the filament to achieve activity and thus avoid the complication of an active protease expressed in the cytoplasm prior to export and polymerization.

We purposely removed the N and C domains in the recombinant construct to avoid the purification and precipitation difficulties due to flagellin polymerization. Nonetheless, when the wild type flagella were purified and assayed for proteolytic activity in the new fibronectin substrate cleavage assay (Supplementary Fig. 9), proteolytic activity was evident in the absence of any bacterial cytotoxicity prior to purification, *i.e.*, flagellin proteolytic activity is apparently effectively chaperoned *in vivo*. In our new data showing modest fibronectin cleavage, it was interesting that the recombinant non-polymerized flagellinolysin enzyme displayed no activity on fibronectin yet did cleave peptide substrates. This suggests that flagellinolysins in their native state, *i.e.* assembled as a component of flagellar filaments, may cleave specific host proteins and that such cleavages are potentiated by structural changes or by processivity efficiencies acquired following assembly onto a stable platform. Thus, flagellinolysin in the context of full-length flagellar filaments can cleave macromolecular protein substrates as may be present in infected animal tissues or biofilms.

A great paper, sure to change the way flagellar biologists and pathogen researchers look at flagellin.

☺ We thank this and all four reviewers for their very positive comments on the impact of our paper.

Page 3 line 20. Consider also including the abbreviation (PAMP) as it might be more recognizable than the full descriptor.

Done as suggested.

Page 4 line 11. Clarify “natural”. Do the authors mean “commonly-studied”?

Done as suggested. By “natural” we mean “non-synthetic” flagellins, but since we feel that inclusion of the term “natural” is unnecessary, it has been removed.

Page 6 line 15. What is PICS?

Done as suggested. We now state on first use P5, L27: “We utilized a high throughput proteomic approach known as Proteomic Identification of protease Cleavage Sites (PICS)^{25,26} to confirm and characterize the metallopeptidase activity of the FliA(H) hypervariable region and peptide substrate specificity. PICS profiles both the prime (P’) and nonprime (P) amino acid preferences of cleavage sites using biological-derived database searchable peptide libraries. PICS confirmed FliA(H)-hypervariable region as an active peptidase, cleaving 391 peptides in a trypsin-generated peptide library, with each peptide ending in either an arginine or a lysine residue (Fig. 4b). To exclude potential library bias, we repeated the PICS analysis using a GluC-generated peptide library, with each peptide ending at glutamate or aspartate residues.

25) Schilling, O. and Overall, C.M. 2008. Proteome-derived Database Searchable Peptide Libraries for Identifying Protease Cleavage Sites. **Nature Biotechnology** **26**, 685-694;

26) Schilling, O., Huesgen, P.F., Barré, O., auf dem Keller, U., and Overall, C.M. 2011. Characterization of the Prime and Non-Prime Active Site Specificities of Proteases by Proteome-derived Peptide Libraries and Tandem Mass Spectrometry. **Nature Protocols** **6**, 111-120

Page 7 line 12. Clarify the point that the organism encoded two different flagellin proteins. Perhaps make this idea more explicit.

Done as suggested. See P7, edits in the paragraph under the header “Flagellinolysins localize to flagellar filament surfaces” now includes: “SDS-PAGE and liquid chromatography-tandem mass spectrometry (LC-MS/MS) verified that flagellar filaments were composed of two different flagellin proteins: the non-proteolytic “structural” flagellin (NCBI accession WP_039229459), which was the dominant band iii, with 82% protein sequence coverage by LC-MS/MS; and the second most abundant protein band ii, identified as the full length proteolytic flagellin protein (annotated as FliA(H) in the database), and having a predicted molecular weight of 60.9 kDa (Fig. 4e) with 79% protein coverage, including the HExxH active site catalytic motif (Supplementary Fig. 7).”

Page 7 line 29. Clarify the origin of the flagellar filaments. Were these filaments sheared from cells and purified?

Done as suggested. Now stated top of P8: “... and full-length proteolytic flagellin from purified sheared filaments (Supplementary Fig. 8).” Also now stated in abstract and legends for pertinent figures, e.g., Figure 4e and Supplementary Figure 4c.

Page 8, line 20. Omit “extremely”. The paper is strong and emphatics are unnecessary.

Done as suggested.

Page 9 line 15-17. Omit.

Done as suggested.

Page 9 line 29. Omit “extreme”. Again the emphatic is unnecessary and it is difficult to qualify the extremeness of horizontal gene transfer.

Done as suggested.

End of discussion. The authors might consider mentioning that their results suggest the possibility that flagellin D3 domains of unknown function, such as that found in E. coli and S. enterica might constitute as yet-unknown enzymatic activities as well.

Done as suggested. This is an interesting idea and is now stated: P10, L23-25:

“It also raises the possibility of enzymatic or other unknown functionality in existing flagellin hypervariable domains of unknown function, *e.g.*, the D3 domain found in *Salmonella* flagellins.”

Reviewer #3:

What are the major claims of the paper?

The authors report the first case of enzymatic function associated with bacterial flagella.

Are they novel and will they be of interest to others in the community and the wider field? Do you feel that the paper will influence thinking in the field?

The authors claim is novel and in my opinion, will be of interest to the scientific community. The primary role of the flagellum is locomotion, but it can also function as a sensory organelle. In addition to these roles, the authors' analysis suggests that flagellum acquired a metallopeptidase gene, which then spread by lateral gene transfer. The peptidase is inserted in the hypervariable region, and it should be facing the exterior of the flagellar filament.

The peptidase acquired by the flagellum carries sequence motifs found in the structure of catalytic domains of bacterial collagenase that Eckhard had previously determined. Evolutional pressure appears to have retained the two features found in the collagenase, i.e. the Ca²⁺ sensors and the catalytic machinery to cleave the peptide bond. However, truncated enzymes lost structural features necessary to recognize and potentially unwind the triple helix of collagen. The peptidase domain was cloned and expressed by the authors and analyzed for its substrate specificity. PICS MS/MS assays of peptides generated by the peptidase do not suggest that it is a collagenase, but it appears to selectively cleave between Tyr and Leu in the AVTY^Y↓LVIA sequence.

The EM clearly shows that the peptidase is located in a regular interval in the surface of intact flagellar filaments isolated from C. haemolyticum. The isolated flagellar filaments exhibited peptidase activity. The PICS MS/MS assays of the isolated filaments indicated Leu was replaced with Gln (GVTY^Y↓QVIN), possibly because of minor structural changes induced by the filamentous arrangement.

The authors also catalogued 85 flagellins with peptidase domains inserted at the hypervariable region. A functional benefit for bacteria possessing peptidase activity in flagellins is unknown at this time. The acquisition of peptidase function may augment its sensory function by generating small peptides and amino acids. Since the peptidase characterized was demonstrated to be highly specific for a certain peptide sequence, it seems likely that it has a specific purpose. Perhaps the motor-driven movement of the flagella combined with proteolytic activity allows the cell to “drill” a hole into a protein-rich matrix during cell invasion.

It reminds me of the uremia of southern India. Because of its danger to the user, it is taught last. Likewise, proteolytic flagella appears to be have been “taught” to select bacteria.

We thank all four reviewers for their very positive comments on our manuscript and interesting ideas on the function of the flagella as a molecular “drill”. We aim to investigate this in future work.

Reviewer #4

The paper by Overall, Doxey and colleagues reports on the identification of a new flagellin family exhibiting proteolytic activity. Lateral gene transfer led to insertion of a zinc-dependent metalloprotease domain into the hypervariable region of flagellin. Bioinformatics revealed this a widespread event between diverse bacterial phyla, including Firmicutes, Beta- and Gammaproteobacteria. Interestingly, some of these protease domain containing flagellins, named flagellinolysins, have additionally been identified in animal pathogens, e.g. in Clostridium haemolyticum. This gives rise to potential functions of flagellinolysins as pathogenic factors. Other possible functions include bacterial defense mechanisms or cell-cell communication.

Good point. We have added these additional two possible functions (pg 10, ln 27-28).

“...including biofilm or host tissue adherence/colonization, virulence, bacterial defense, or extracellular cell-cell communication.”

For biochemical and functional characterization the authors produced recombinant flagellinolysin from C. haemolyticum, called FliA(H), and investigated its catalytic activity employing a proteomics approach for the identification cleavage site specificity (PICS). Interestingly, the specificity turned out to be MMP-like, with preference for leucine in P1', which was additionally validated in single peptide cleavage assays. As control EDTA as metalloprotease inhibitor was used and a catalytically inactive mutant of FliA(H) was produced. For more physiological assays, the authors isolated intact flagellar filaments from anaerobically cultured C. haemolyticum and again proofed catalytic activity employing PICS validating the results observed with recombinant enzyme.

The discovery of proteolytically active flagellar filaments is a striking finding important for a wide multidisciplinary field. It provides a new fascinating research aspect with possible impact on the understanding of host-microbiome interaction. All experiments and data presented herein are of high quality and support the discovery of flagellinolysins as active proteases.

We thank all 4 reviewers for their very positive comments on our work and its potential impact across many disciplines.

However, one additional experimental approach should be considered prior publication. As nicely stated by the authors “With ~20,000 flagellin copies per ~10 μm flagella this assembles the largest proteolytic complex known to date.” and with regard to such high abundance of enzymatic activity on C. haemolyticum, I suggest to analyze potential substrates in the host proteome. C. haemolyticum can infect man and animals, in some cases through the intestinal epithelium. The group of Dr. Overall developed a very effective technique for the identification of protease substrates. The N terminomics approach TAILS will provide evidence, if the proteolytic activity of FliA(H) specifically cleaves cell surface or secreted proteins produced by cultured intestinal cells. This would further support the importance of this kind of protease and would strengthen the biological part of this study.

The reviewer raises an intriguing hypothesis about possible biological substrates of C.

haemolyticum flagellinolysin. Indeed, we considered TAILS some time ago presubmission. However, lack of the diglycine active site motif and the high homology of C. *haemolyticum* flagellinolysin with other bacterial flagellinolysins of mostly non-invasive, nonpathogenic bacteria, e.g., in marine environments/biofilms, suggests flagellinolysin may be primarily function as a peptidase and not an endoprotease. Thus, an extensive protease screen such as TAILS would likely be disappointing and further delay resubmission. Instead, we selected and tested three potential substrates in the host proteome based on the following criteria:

1. As suggested by the reviewer, flagellinolysin may cleave abundant cell-surface associated or secreted extracellular matrix proteins in tissues of C. *haemolyticum* infection (i.e., gut epithelial and subjacent cells, liver). Thus, native and denatured (gelatin) type I collagen, fibronectin, and the mucous membrane immunoglobulin A (IgA) were selected as candidates of potential physiological relevance.
2. Known targets of detectable homologs (i.e., family M26) was emphasized by another positive reviewer, suggesting IgA as a possible substrate.

3. Known targets of proteases with similar PICS profiles (MMPs with similar profiles are known to cleave fibronectin and show activity against gelatin). Unpublished data from the Holyoak lab also revealed a PICS profile that is similar to flagellinolysin for an immunoglobulin A1 protease.

Based on these criteria, we selected bovine fibronectin (satisfies criteria 1 and 3), rat type I collagen (native and denatured) and bovine gelatin (satisfies 1 and 2), and human IgA1 (satisfies 1, 2, and 3). We then tested all candidate substrates using both the recombinant flagellinolysin enzyme and purified flagellar filaments from *C. haemolyticum*.

We observed modest, yet clear degradation of fibronectin using intact flagellar filaments (see **new Supplementary Figure 9**), but not by recombinant non-polymerized flagellinolysin enzyme. This suggests that flagellinolysins in their native state (assembled as component of flagellar filaments) may cleave specific host proteins and that such cleavages are potentiated by structural changes or by processivity efficiencies acquired following assembly onto a stable platform. Fibronectin hydrolysis was inhibited completely by EDTA and no contaminating proteases in the preparations were identified by mass-spectrometry as discussed in the text and in Supplementary Fig. 7. This new fibronectin cleavage data supports the reviewer's hypothesis that flagellinolysins may target abundant cell surface or extracellular matrix proteins in the host gut epithelium or liver. Notably, fibronectin is also a known substrate of MMPs, which we identified in the manuscript as sharing a similar PICS profile. We thank the reviewer for suggesting this direction, and have added this information to the manuscript at the end of Results (P 8, L 21-end).

Despite good efforts and different conditions, protease assays were negative on native and denatured type I collagen and IgA1.

“Finally, we observed minor flagellar filament-associated autocatalytic activity and modest cleavage of bovine native fibronectin, which was also inhibited by EDTA (Supplementary Fig. 9). As plasma fibronectin is a major secretory protein of liver hepatocytes and cellular fibronectin is an important cell adhesive glycoprotein in connective tissues, fibronectin degradation and depolymerization of the multimeric 440-kDa crosslinked form may facilitate *C. haemolyticum* colonization, tissue spreading and necrosis. Taken together, these results demonstrated that *C. haemolyticum* proteolytic flagellins are abundant constituents of assembled flagella, confer peptidase activity to intact filaments, and degrade biological targets of potential physiological relevance to *C. haemolyticum* virulence.”

REVIEWERS' COMMENTS:

Reviewer #1 (Remarks to the Author):

The authors have made all the necessary changes, including extra experimental work, requested by myself and the other reviewers and I have no further changes to suggest.

Reviewer #2 (Remarks to the Author):

The authors addressed my concerns. Cool story.

Reviewer #4 (Remarks to the Author):

Substrates define proteases' roles. Therefore, I appreciate that the authors considered investigation of extracellular matrix proteins as putative substrates. However, the authors argued against the TAILS approach, as they assumed that flagellinolysin is rather a peptidase than an endoproteinase. Now they show that flagellinolysin is capable of degrading fibronectin, which clearly demonstrates endoproteinase activity. Most interestingly, cleavage of fibronectin was only observed using intact flagellar filaments, but not by recombinant flagellinolysin. This observation alone would deserve deeper investigation on the biochemical characteristics of the protease, ideally in combination with an N-terminomics approach based on a suitable cell line. Nevertheless, the identification of this new class of bacterial proteases and basic proteolytic characterization is of broad scientific interest and can now be accepted for publication. Hopefully, the authors will follow-up this topic and provide deeper insight into the function of these enzymes in the future.